# Adaptive Prototype Learning: Unlocking Intrinsic Features for Texture Recognition

## Abstract

State-of-the-art texture recognition models often rely on cumbersome external memory banks and complex training pipelines. We challenge this paradigm by proposing a simple yet powerful alternative: learning from the rich intrinsic patterns within each image itself. We introduce STP-Former (Simple Texture Prototype Transformer), an architecture that dynamically distills a compact set of intrinsic prototypes for each input sample. A lightweight cross-attention module, the Texture Prototype Extractor (TPE), learns to identify and aggregate an image's most representative texture primitives on-the-fly. These adaptive prototypes, inherently aligned with the input's context, form a powerful basis for robust classification.Our contributions are twofold. First, we propose a decoupled two-stage training strategy where the TPE is pre-trained using a self-supervised objective to capture fundamental texture representations before a classifier is fine-tuned. Second, to endow the learned feature space with a robust geometric structure, we introduce a novel Supervised Topological Loss. Grounded in persistent homology, this objective directly optimizes for intra-class compactness and inter-class separation, pushing the boundaries of discriminability. This synergistic framework yields a remarkable performance leap; on the challenging DTD benchmark, STP-Former improves accuracy from 79% to over 86%. Our work demonstrates that an adaptive, self-contained approach provides a more effective and efficient paradigm for texture recognition.

## 1 Introduction

Texture, as a fundamental visual attribute, encapsulates the spatial organization of basic elements within texture-rich images, serving as a vital representation of the underlying microstructure in natural scenes Liu et al. (2019). Textured regions are typically characterized by repetitive patterns with inherent variability, making them essential pre-attentive visual cues for comprehending natural scenes. This unique property has enabled a wide range of applications, including medical image analysis Peikari et al. (2015), content-based image retrieval, and material classification Liu et al. (2019).

For decades, handcrafted texture descriptors formed the basis of classical material and texture recognition methods. Techniques such as Gray-Level Co-occurrence Matrices (GLCM) Haralick et al. (1973), Local Binary Patterns (LBP) Kylberg & Sintorn (2013), and Gabor Filters Idrissa & Acheroy (2002) were widely utilized. Further advancements introduced aggregation-based approaches like Bag of Words (BoW) and Vector of Locally Aggregated Descriptors (VLAD) Jégou et al. (2010).

With the rise of deep learning, Convolutional Neural Networks (CNNs) and more recently Vision Transformers (ViTs) Dosovitskiy et al. (2020) have become the dominant framework. Methods such as FV-CNN Liu et al. (2019), DeepTEN Zhang et al. (2017), and DSRNet Zhai et al. (2020) leverage deep representations to extract texture features effectively. More recent approaches, including CLASSNet Chen et al. (2021) and FENet Xu et al. (2021), have incorporated multi-scale fractal analysis to better adapt to spatial distributions. A parallel line of work has focused on leveraging large, external memory banks of features, such as in PatchCore Roth et al. (2022), to compare test samples against a comprehensive library of normal patterns.

Despite their success, these state-of-the-art methods share a fundamental limitation: a reliance on *extrinsic*, pre-compiled knowledge derived from a training set. Whether it's a learned statistical dis-

tribution or a static memory bank, these methods face an inherent **feature misalignment problem**. Variations in pose, illumination, and scale in a test image often lead to a domain shift, causing its features to misalign with the static representations learned from the training set. This misalignment acts as a performance ceiling, as we formalize in Section 3.

To address this fundamental challenge, we propose a paradigm shift. Inspired by the observation that even a single texture image contains rich, repetitive patterns sufficient for self-characterization Luo et al. (2025), and building upon recent advances in adaptive prototype learning for various vision tasks Li et al. (2021); Heidari et al. (2024); Ma et al. (2024), we introduce a simple yet remarkably effective framework called **STP-Former (Simple Texture Prototype Transformer)**. Our approach abandons the reliance on external knowledge bases and instead learns to extract a compact set of **Intrinsic Prototypes (IPs)** dynamically from each input image. This is our first major contribution: a decoupled two-stage training strategy where a **Texture Prototype Extractor (TPE)** is first trained with a self-supervised objective to distill representative texture primitives. Subsequently, a simple classifier is trained using these powerful, highly-aligned intrinsic features. While this approach effectively solves the feature misalignment problem, it raises a subsequent question: how can we best structure the geometry of this new adaptive feature space to maximize class separability?

To answer this, we introduce our second major contribution, which moves beyond conventional losses. We propose to explicitly sculpt the feature manifold using principles from Topological Data Analysis (TDA). While TDA has often been used as a regularizer to preserve existing data structures, we employ it as a direct, supervised optimization objective to actively construct a more discriminative space. We put this concept into practice through a novel Supervised Topological Loss, an objective grounded in persistent homology that directly enforces intra-class compactness and inter-class separation. The synergy between adaptive intrinsic prototypes and a topologically structured feature space creates a simple yet powerful framework that achieves a new state-of-the-art, improving accuracy on the challenging GTOS dataset by over 10 percentage points.

## 2 RELATED WORK

### 2.1 DEEP LEARNING FOR TEXTURE RECOGNITION

The advent of deep learning has revolutionized texture recognition. Early works adapted pre-trained CNNs, often combining them with traditional encoding methods like Fisher Vectors (FV-CNN) Cimpoi et al. (2015). To enable end-to-end training, methods like DeepTEN Zhang et al. (2017) integrated dictionary learning and residual encoding directly into the network. Subsequent works have focused on capturing the complex statistical properties and spatial dependencies inherent in textures. For instance, CLASSNet Chen et al. (2021) and FENet Xu et al. (2021) successfully applied fractal analysis to model the statistical self-similarity across feature layers. DSRNet Zhai et al. (2020) and MPAP Zhai et al. (2023) explored the spatial dependency of texture primitives and the semantic relationship between texture attributes. While powerful, these methods primarily focus on designing sophisticated modules to better model the distribution of features learned from a training set, without explicitly addressing the potential misalignment with test data.

### 2.2 TOPOLOGICAL DATA ANALYSIS IN DEEP LEARNING

Topological Data Analysis (TDA), particularly through its key tool of persistent homology Dey & Wang (2022), has emerged as a powerful method for analyzing high-dimensional data structures in machine learning. Its applications are broad, ranging from enforcing topological priors in computer vision tasks like image segmentation Hu et al. (2019); Clough et al. (2020) to enhancing the expressiveness of graph neural networks Yan et al. (2021); Immonen et al. (2023). A significant line of work focuses on using TDA as a regularizer to learn or preserve the topological structure of feature spaces, especially in representation and generative learning contexts Moor et al. (2020); Barannikov et al. (2022); Mishra et al. (2024). While these methods typically aim to maintain an assumed or existing data topology, our work takes a different approach. We employ TDA not as a preservative regularizer, but as a direct, supervised optimization objective. Our Supervised Topological Loss actively constructs a new, geometrically structured feature space by explicitly enforcing intra-class compactness and inter-class separation, thereby directly enhancing the model's discriminative capabilities for the classification task.

## 3 PRELIMINARY: THE PROBLEM OF FEATURE MISALIGNMENT IN TEXTURE RECOGNITION

A dominant paradigm in modern texture recognition, especially in methods striving for state-of-the-art performance, is based on metric learning against a set of pre-compiled representations. Let $\mathcal{X}_{\text{train}}$ be the training set and $\Phi : \mathcal{I} \to \mathbb{R}^{N \times D}$ be a deep feature extractor that maps an image $x \in \mathcal{I}$ to a set of $N$ patch features. A memory bank or a set of class prototypes for each class $c$ is constructed from the training data, denoted as $\mathcal{M}_c = \{\mathbf{p}_{ci}\}_{i=1}^{K} \subset \mathbb{R}^D$, where each prototype $\mathbf{p}_{ci}$ is derived from $\{\Phi(x) \mid x \in \mathcal{X}_{\text{train}}, y(x) = c\}$.

The classification of a test image $x_{\text{test}}$ is then determined by a decision function $g$ that measures the similarity or distance between its features $\Phi(x_{\text{test}})$ and the prototypes of each class:

$$\hat{y} = \arg\min_c \mathcal{D}(\Phi(x_{\text{test}}), \mathcal{M}_c) \tag{1}$$

where $\mathcal{D}$ is a distance metric (e.g., Euclidean distance to the nearest prototype). This formulation implicitly assumes that the feature distribution of test images for a class $c$, $P_{\text{test}}(\Phi(x) \mid y(x) = c)$, is well-aligned with the distribution represented by the training prototypes $\mathcal{M}_c$.

However, in real-world scenarios ("in the wild"), this assumption is frequently violated. Let $\mathcal{T}$ be a set of transformations (e.g., changes in illumination, scale, viewpoint) that preserve the semantic texture class. For a test image $x'_{\text{test}} = T(x_{\text{test}})$ where $T \in \mathcal{T}$, its feature representation $\Phi(x'_{\text{test}})$ may undergo a significant shift in the feature space. We define this as the **Feature Misalignment Problem**: there exists a transformation $T$ such that even if $y(x'_{\text{test}}) = y(x_{\text{test}}) = c$, the feature distance increases significantly:

$$\mathcal{D}(\Phi(T(x_{\text{test}})), \mathcal{M}_c) \gg \mathcal{D}(\Phi(x_{\text{test}}), \mathcal{M}_c) \tag{2}$$

This can lead to a situation where the misaligned feature $\Phi(T(x_{\text{test}}))$ becomes closer to the prototype set of an incorrect class $c' \neq c$:

$$\mathcal{D}(\Phi(T(x_{\text{test}})), \mathcal{M}_{c'}) < \mathcal{D}(\Phi(T(x_{\text{test}})), \mathcal{M}_c) \tag{3}$$

This misalignment creates a fundamental performance bottleneck for methods reliant on static, extrinsic knowledge from the training set. It necessitates a new approach that can generate representations inherently aligned with the context of each individual test image. Our work directly addresses this challenge by proposing a framework that extracts prototypes *intrinsically* from each image, ensuring perfect alignment by design.

## 4 METHODOLOGY

Our approach, the **Simple Texture Prototype Transformer (STP-Former)**, is designed to overcome the feature misalignment problem inherent in methods relying on static, extrinsic knowledge. We achieve this by dynamically extracting a set of intrinsic prototypes from each test image. The framework is realized in two variants: a foundational **STP-Former** and an enhanced version, **STP-Former+**, which incorporates a novel topological loss to explicitly structure the feature space. Both models are trained using a decoupled two-stage strategy that first learns a powerful, self-supervised texture representation and then fine-tunes a classifier on top of it.

### 4.1 CORE ARCHITECTURE

The architecture is shared between both STP-Former and STP-Former+ and consists of three main components: a backbone feature extractor, a bottleneck module for feature refinement, and our core Texture Prototype Extractor (TPE).

**Backbone and Feature Fusion.** We employ the DINOv2 model Oquab et al. (2023) with a Vision Transformer (ViT-B/14) architecture as our backbone, leveraging its powerful representations pre-trained on a large-scale dataset. For a given input image $x \in \mathbb{R}^{H \times W \times 3}$ (with $H = W = 518$), the ViT backbone produces a sequence of patch tokens. To capture a rich, multi-level representation of texture, we extract features from intermediate layers 2 through 9. These features are then fused via element-wise averaging to produce a single, comprehensive set of patch features $\mathbf{F}_{\text{raw}} \in \mathbb{R}^{N \times D}$, where $N = 1369$ is the number of patches and $D = 768$ is the feature dimension.

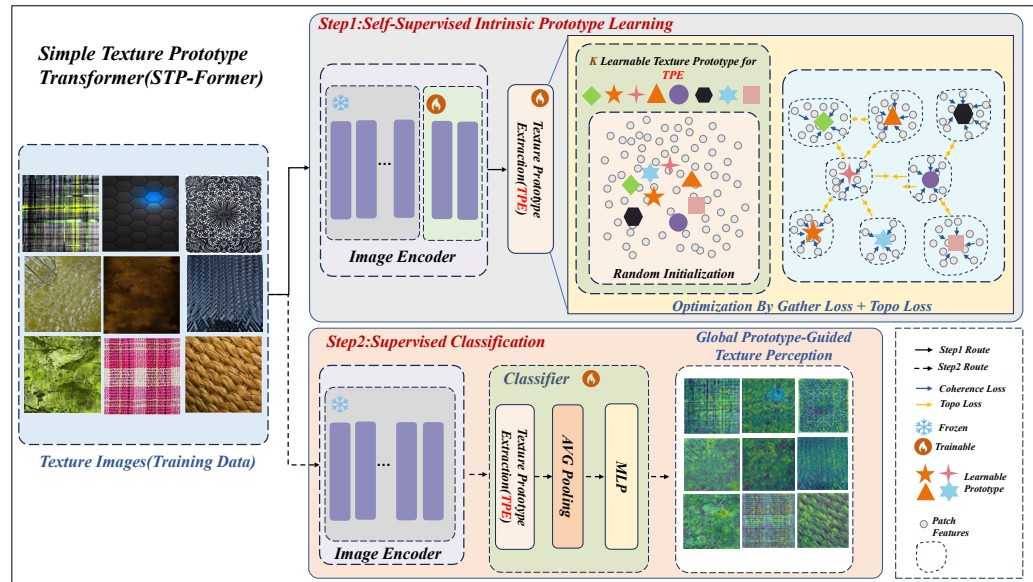

Figure 1: **The overall architecture of our Simple Texture Prototype Transformer (STP-Former).** Our framework is based on a decoupled two-stage training strategy. **Stage 1: Self-Supervised TPE Training.** An image encoder extracts patch features from training images. A set of $K$ learnable texture prototype queries are optimized via a self-supervised objective (**Gather Loss**), which ensures the resulting prototypes are representative of the patch features. **Stage 2: Supervised Classifier Training.** The pre-trained and frozen TPE is used to extract intrinsic prototypes from images. These are aggregated and fed into a simple, trainable classifier. For our enhanced model, STP-Former+, a **Supervised Topological Loss** is added in this stage to regularize the feature space geometry, enforcing class separation.

**Texture Prototype Extractor (TPE).**  The TPE is the central component responsible for distilling intrinsic prototypes. It comprises two sub-modules:

(i) **Bottleneck MLP:** The fused patch features $\mathbf{F}_{\text{raw}}$ are first passed through a simple MLP with a bottleneck structure ($D \rightarrow 4D \rightarrow D$) to refine and transform the features into a more discriminative space, resulting in $\mathbf{F}_{\text{refined}} \in \mathbb{R}^{N \times D}$.

(ii) **Intrinsic Prototype Aggregator:** The core of our method lies in this module. We initialize a set of $K = 16$ learnable vectors, $\mathbf{P}_{\text{query}} \in \mathbb{R}^{K \times D}$, which serve as **learnable texture prototype queries**. These queries are shared across all images. For each image, the aggregator uses a multi-head cross-attention mechanism where $\mathbf{P}_{\text{query}}$ acts as the query and the image's refined patch features $\mathbf{F}_{\text{refined}}$ serve as the key and value. This allows the queries to "attend" to the most salient texture primitives within the image and aggregate them into a compact set of $K$ **Intrinsic Prototypes (IPs)**, $\mathbf{P}_{\text{out}} \in \mathbb{R}^{K \times D}$.

The complete data flow is as follows:

$$\text{Image} \xrightarrow{\text{DINOv2}} \mathbf{F}_{\text{raw}} \xrightarrow{\text{MLP}} \mathbf{F}_{\text{refined}} \xrightarrow[\text{Cross-Attention}]{\mathbf{P}_{\text{query}}} \mathbf{P}_{\text{out}}$$

## 4.2 DECOUPLED TWO-STAGE TRAINING STRATEGY

We adopt a two-stage training strategy to ensure that the TPE learns a robust and general representation of texture primitives before being adapted for the specific classification task. The overall process is detailed in Algorithm 1.

#### 4.2.1 STAGE 1: SELF-SUPERVISED INTRINSIC PROTOTYPE LEARNING

The goal of this stage is to train the TPE to generate prototypes that faithfully represent the texture patterns within an image, independent of class labels.

**Objective and Loss Function.** To ensure the extracted intrinsic prototypes $\mathbf{P}_{\text{out}} = \{\mathbf{p}_k\}_{k=1}^{K}$ are meaningful representatives of the image's patch features $\mathbf{F}_{\text{refined}} = \{\mathbf{f}_n\}_{n=1}^{N}$, we employ a self-supervised objective. The core idea is to compel the prototypes to collectively cover the feature manifold of the patches. This is achieved by minimizing the average distance from each patch feature to its nearest prototype in the cosine similarity space, effectively ensuring that every patch feature is well-represented by the prototype set. We refer to this objective as the **Gather Loss** Luo et al. (2025), which is formally defined as:

$$\mathcal{L}_{\text{gather}} = \frac{1}{N} \sum_{n=1}^{N} \left( 1 - \max_{k \in \{1, \ldots, K\}} \frac{\mathbf{f}_n \cdot \mathbf{p}_k}{\|\mathbf{f}_n\| \|\mathbf{p}_k\|} \right) \tag{4}$$

Minimizing this loss forces the learnable queries $\mathbf{P}_{\text{query}}$ and the bottleneck MLP to learn a process that distills the most representative texture features from any given image into the $K$ prototypes.

**Training Details.** During this stage, we freeze the initial 8 layers of the DINOv2 backbone and fine-tune the final 4 layers. The trainable parameters include these top layers of the backbone, the bottleneck MLP, and the INP Aggregator (including the prototype queries $\mathbf{P}_{\text{query}}$). We use the AdamW optimizer with a learning rate of $1 \times 10^{-3}$ and a weight decay of $1 \times 10^{-4}$, training for 100 epochs with a batch size of 16.

#### 4.2.2 STAGE 2: SUPERVISED CLASSIFICATION

In the second stage, we freeze the pre-trained TPE and train a classifier to map the extracted intrinsic prototypes to their corresponding class labels.

**Feature Aggregation and Classification.** For each image, the frozen TPE outputs a set of $K$ intrinsic prototypes $\mathbf{P}_{\text{out}} \in \mathbb{R}^{K \times D}$. These prototypes are aggregated into a single global feature vector $\mathbf{z} \in \mathbb{R}^{D}$ via global average pooling:

$$\mathbf{z} = \frac{1}{K} \sum_{k=1}^{K} \mathbf{p}_k \tag{5}$$

This global feature $\mathbf{z}$ is then passed to a lightweight classification head, consisting of Layer Normalization, a Dropout layer ($p = 0.1$), and a final linear layer that maps $\mathbf{z}$ to the class logits.

**Training STP-Former (Base Version).** The base model is trained by minimizing the standard **Cross-Entropy Loss** ($\mathcal{L}_{\text{CE}}$) between the predicted logits and the ground-truth labels. To maintain the quality of the pre-trained feature extractor, we use a differential learning rate scheme: the TPE components are fine-tuned with a small learning rate ($1 \times 10^{-5}$), while the new classification head is trained with a larger learning rate ($1 \times 10^{-4}$). The model is trained for 50 epochs.

### 4.3 STP-FORMER+: REGULARIZATION WITH TOPOLOGICAL LOSS

The key innovation of STP-Former+ is the introduction of a novel **Supervised Topological Loss** during Stage 2, designed to explicitly regularize the geometric structure of the feature space. This approach enhances the model's generalization by enforcing intra-class compactness and inter-class separation from a topological perspective.

**Theoretical Foundation.** Our method leverages concepts from Topological Data Analysis (TDA), specifically **0-dimensional Persistent Homology** ($H_0$). For a mini-batch of global features, we construct a **Vietoris-Rips filtration**, which is a sequence of simplicial complexes built by progressively adding edges between points in increasing order of their distance. $H_0$ persistent homology tracks the birth and death of connected components throughout this filtration. A component is "born" at the beginning (each point is its own component) and "dies" when it merges with an older, existing component. The distance at which this merge occurs is recorded as the component's **death time**.

---

**Algorithm 1** Training Procedure for STP-Former

---

1: **Input:** Training dataset $\mathcal{D}$, DINOv2 backbone $\Phi$, num prototypes $K$, learning rates $\eta_1, \eta_2$, epochs $E_1, E_2$.
2: **Initialize:** TPE parameters $\theta_{\text{TPE}}$ (Bottleneck MLP $B$, Prototype Queries $\mathbf{P}_{\text{query}}$), Classifier parameters $\theta_C$.

3:                       ▷ **— Stage 1: Self-Supervised TPE Training —**
4: **for** epoch = 1 to $E_1$ **do**
5:      **for** image batch $\mathbf{x} \in \mathcal{D}$ **do**
6:          Extract multi-layer patch features: $\mathbf{F}_{\text{raw}} \leftarrow \Phi(\mathbf{x})$
7:          Refine patch features: $\mathbf{F}_{\text{refined}} \leftarrow B(\mathbf{F}_{\text{raw}})$
8:          Aggregate intrinsic prototypes: $\mathbf{P}_{\text{out}} \leftarrow \text{CrossAttention}(\mathbf{P}_{\text{query}}, \mathbf{F}_{\text{refined}})$
9:          Compute Gather Loss using Eq. 4: $\mathcal{L}_{\text{gather}} \leftarrow \text{ComputeLoss}(\mathbf{F}_{\text{refined}}, \mathbf{P}_{\text{out}})$
10:         Update trainable parameters of $\Phi$ and $\theta_{\text{TPE}}$ using $\nabla_{\Phi, \theta_{\text{TPE}}} \mathcal{L}_{\text{gather}}$ with learning rate $\eta_1$.
11:      **end for**
12: **end for**

13:                      ▷ **— Stage 2: Supervised Classifier Training —**
14: Freeze parameters of $\Phi$ and TPE.
15: **for** epoch = 1 to $E_2$ **do**
16:      **for** (image batch $\mathbf{x}$, label batch $\mathbf{y}$) $\in \mathcal{D}$ **do**
17:                      ▷ Forward pass with no gradient through the extractor
18:          $\mathbf{F}_{\text{raw}} \leftarrow \Phi(\mathbf{x});$     $\mathbf{F}_{\text{refined}} \leftarrow B(\mathbf{F}_{\text{raw}})$
19:          $\mathbf{P}_{\text{out}} \leftarrow \text{CrossAttention}(\mathbf{P}_{\text{query}}, \mathbf{F}_{\text{refined}})$
20:          Aggregate to global feature: $\mathbf{z} \leftarrow \text{AveragePool}(\mathbf{P}_{\text{out}})$
21:          Predict logits: logits $\leftarrow C(\mathbf{z})$
22:          Compute Cross-Entropy Loss: $\mathcal{L}_{\text{CE}} \leftarrow \text{CrossEntropy}(\text{logits}, \mathbf{y})$
23:          Update $\theta_C$ using $\nabla_{\theta_C} \mathcal{L}_{\text{CE}}$ with learning rate $\eta_2$.
24:          (Optional) Fine-tune $\theta_{\text{TPE}}$ with learning rate $\eta_2/10$.
25:      **end for**
26: **end for**
27: **Output:** Trained STP-Former model with parameters $\theta_{\text{TPE}}$ and $\theta_C$.

---

**Topological Loss Formulation.** For a mini-batch of $B$ global features $\{\mathbf{z}_i\}_{i=1}^{B}$ with corresponding labels $\{y_i\}_{i=1}^{B}$, we first compute the pairwise Euclidean distance matrix $D_{ij} = d(\mathbf{z}_i, \mathbf{z}_j)$. We then apply the persistent homology algorithm to identify the set of **critical edges**—pairs of indices $(i, j)$ whose corresponding edge in the filtration causes two previously disconnected components to merge. The death time of such an event is precisely the distance $d(\mathbf{z}_i, \mathbf{z}_j)$. Our topological loss, $\mathcal{L}_{\text{topo}}$, is composed of two terms that operate on these critical edges:

- **Intra-Class Compactness Loss ($\mathcal{L}_{\text{intra}}$):** This term encourages features from the same class to be close. We sum the distances of all critical edges that connect points $(i, j)$ belonging to the *same class*. Minimizing this term forces same-class samples to merge early in the filtration, promoting a compact class cluster.

$$\mathcal{L}_{\text{intra}} = \sum_{\substack{(i,j) \in E_{\text{crit}} \\ y_i = y_j}} d(\mathbf{z}_i, \mathbf{z}_j) \tag{6}$$

- **Inter-Class Separation Loss ($\mathcal{L}_{\text{inter}}$):** This term encourages features from different classes to be far apart. We sum the *negative* distances of all critical edges connecting points $(i, j)$ from *different classes*. Minimizing this term is equivalent to maximizing their merge distance, pushing them to connect as late as possible in the filtration and thus promoting separation between class clusters.

$$\mathcal{L}_{\text{inter}} = - \sum_{\substack{(i,j) \in E_{\text{crit}} \\ y_i \neq y_j}} d(\mathbf{z}_i, \mathbf{z}_j) \tag{7}$$

Here, $E_{\text{crit}}$ denotes the set of all critical pairs identified by the $H_0$ algorithm. The total topological loss is a weighted sum of these two components: $\mathcal{L}_{\text{topo}} = \mathcal{L}_{\text{intra}} + \lambda_{\text{inter}} \mathcal{L}_{\text{inter}}$, where we set $\lambda_{\text{inter}} = 0.5$ based on empirical validation.

**Final Objective for STP-Former+.** The complete loss function for training STP-Former+ in Stage 2 combines the standard classification objective with our topological regularizer:

$$\mathcal{L}_{\text{total}} = \mathcal{L}_{\text{CE}} + \lambda_{\text{topo}} \mathcal{L}_{\text{topo}} \tag{8}$$

where $\lambda_{\text{topo}} = 0.1$ is a hyperparameter balancing the two loss terms. This composite loss trains the classifier not only to be accurate but also to produce a feature space with a robust and well-separated geometric structure, enhancing generalization. The training setup (epochs, learning rates) remains the same as the base STP-Former.

## 5 EXPERIMENTS

### 5.1 EXPERIMENTAL SETTING

**Datasets.** The proposed method is evaluated on six widely-used benchmark datasets. The Describable Textures Database (DTD) Cimpoi et al. (2014) comprises 47 texture categories, each containing 120 images, with ten predefined splits for training, validation, and testing. The Flickr Material Dataset (FMD) Sharan et al. (2013) consists of ten material categories and is a standard benchmark for material classification. The Materials in Context Database (MINC) Bell et al. (2015) includes 23 material classes, with 2500 images per class, and provides five training/testing splits. The Fabrics dataset Kampouris et al. (2016) serves as a publicly available resource for fine-grained material classification. Ground Terrain in Outdoor Scenes (GTOS) Xue et al. (2017) consists of 40 outdoor ground material classes, with a predefined training/testing split. Finally, the KTH-TIPS2b Caputo et al. (2005) dataset includes texture-rich images from 11 material categories, captured under various conditions to simulate realistic scenarios.

**Implementation Details.** For a fair comparison, we implement our methods with two different backbones: a standard ResNet-50 and the more powerful DINOv2-ViT-B/14. All experiments are conducted using the PyTorch framework on a single NVIDIA A100 GPU. Images are resized to $518 \times 518$. We follow the two-stage training strategy detailed in Section 4. In Stage 1, the TPE is trained for 100 epochs using the AdamW optimizer with a learning rate of $1 \times 10^{-3}$. In Stage 2, the classifier is trained for 50 epochs with a differential learning rate scheme: $1 \times 10^{-4}$ for the classifier head and $1 \times 10^{-5}$ for fine-tuning the TPE. The batch size is set to 16 for all experiments. For our STP-Former+ model, the topological loss hyperparameters are set to $\lambda_{\text{topo}} = 0.1$ and $\lambda_{\text{inter}} = 0.5$.

### 5.2 COMPARISON WITH STATE-OF-THE-ART METHODS

As shown in Table 1, our proposed methods, STP-Former and STP-Former+, achieve new state-of-the-art results across all six challenging texture recognition benchmarks.

Even when using a standard ResNet backbone, our approach demonstrates a distinct advantage over previous methods. **STP-Former+ (ResNet)** surpasses the strong baseline GraphTEN Peng et al. (2025) on 5 out of 6 datasets, with notable gains on GTOS (88.1% vs. 86.8%) and KTH (89.2% vs. 87.4%), showcasing the general effectiveness of our intrinsic prototype extraction and topological regularization framework.

The performance is further amplified when leveraging the powerful DINOv2 backbone. Our **STP-Former+ (DINOv2)** model establishes a new SOTA by a significant margin across all datasets. The most remarkable improvements are observed on the DTD dataset, where our model achieves **86.1%** accuracy, a substantial **7.0%** absolute improvement over the previous best, GraphTEN. This highlights our model's superior ability to handle the diverse and complex patterns in DTD. Similarly, on GTOS, FMD, and KTH, our model pushes the state-of-the-art to **88.7%**, **90.3%**, and **90.0%** respectively.

Furthermore, a consistent performance gap is observed between our base model, STP-Former, and the enhanced STP-Former+. The introduction of the Supervised Topological Loss consistently yields

an accuracy boost of 1-2% across most datasets and backbones (e.g., 84.47% to 86.1% on DTD with DINOv2). This empirically validates the effectiveness of our topological loss in structuring the feature space for better class separability. These results collectively demonstrate that our proposed paradigm of combining dynamic intrinsic prototype extraction with explicit geometric regularization sets a new standard for texture recognition.

Table 1: Performance comparison of different methods in terms of classification accuracy (%). The best results are in **bold**, and the second best are underlined.

| Method | DTD | | MINC | | FMD | | Fabrics | | GTOS | | KTH | |
|---|---|---|---|---|---|---|---|---|---|---|---|---|
| | mean | std | mean | std | mean | std | mean | std | mean | std | mean | std |
| FC-CNN(CVPR15)Cimpoi et al. (2015) | 62.9 | 0.8 | 60.4 | 0.5 | 77.5 | 1.8 | 57.9 | 0.6 | 68.5 | 0.6 | 81.8 | 2.5 |
| FV-CNN(CVPR15)Cimpoi et al. (2015) | 72.3 | 1.0 | 69.8 | 0.5 | 79.8 | 1.8 | 66.5 | 0.9 | 77.1 | 0.6 | 75.4 | 1.5 |
| BCNN(CVPR16)Lin & Maji (2016) | 69.6 | 0.7 | 67.1 | 1.1 | 77.8 | 1.9 | 65.6 | - | 78.7 | 0.3 | 75.1 | 2.8 |
| Deep-TEN(CVPR17)Zhang et al. (2017) | 69.6 | 0.5 | 81.3 | 0.7 | 80.2 | 0.9 | 75.2 | 0.7 | 84.5 | 0.4 | 82.0 | 3.3 |
| DEP(CVPR18)Xue et al. (2018) | 73.2 | 0.5 | 82.0 | 0.7 | 80.7 | 0.7 | 74.3 | 1.2 | - | - | 82.4 | 3.5 |
| MAPNet(ICCV19)Zhai et al. (2019) | 76.1 | 0.6 | - | - | 85.2 | 0.7 | - | - | 84.7 | 2.2 | 84.5 | 1.3 |
| DSRNet(CVPR20)Zhai et al. (2020) | 77.6 | 0.6 | - | - | 86.0 | 0.8 | - | - | 85.3 | 2.0 | 85.9 | 1.3 |
| HistNet(PR21)Peeples et al. (2020) | 72.0 | 1.2 | 82.4 | 0.3 | - | - | - | - | - | - | - | - |
| FENet(NeurIPS21)Xu et al. (2021) | 74.2 | 0.1 | 83.9 | 0.1 | 86.7 | 0.2 | - | - | 85.7 | 0.1 | 88.2 | 0.2 |
| CLASSNet(CVPR21)Chen et al. (2021) | 74.0 | 0.5 | 84.0 | 0.6 | 86.2 | 0.9 | - | - | 85.6 | 2.2 | 87.7 | 1.3 |
| MPAP(TPAMI23)Zhai et al. (2023) | 78.0 | 0.5 | 82.5 | 0.1 | 87.6 | 0.9 | - | - | 86.1 | 1.8 | 87.9 | 1.5 |
| GraphTEN(ICME2025)Peng et al. (2025) | 79.1 | 0.6 | 85.2 | 0.3 | 87.7 | 1.2 | 80.7 | 0.6 | 86.8 | 2.5 | 87.4 | 1.6 |
| *Our methods with ResNet backbone* | | | | | | | | | | | | |
| STP-Former (ResNet) | 80.4 | 0.5 | 86.1 | 0.4 | 88.6 | 0.8 | 81.5 | 0.5 | 87.2 | 1.9 | 88.3 | 1.2 |
| STP-Former+ (ResNet) | 81.7 | 0.4 | 86.9 | 0.3 | 89.5 | 0.7 | 82.3 | 0.4 | 88.1 | 1.5 | 89.2 | 1.0 |
| *Our methods with DINOv2 backbone* | | | | | | | | | | | | |
| STP-Former (DINOv2) | 84.47 | 0.3 | 87.5 | 0.3 | 89.2 | 0.6 | 83.5 | 0.4 | 87.7 | 0.6 | 88.9 | 1.1 |
| STP-Former+ (DINOv2) | **86.1** | 0.2 | **88.6** | 0.2 | **90.3** | 0.5 | **84.7** | 0.3 | **88.7** | 1.7 | **90.0** | 0.8 |

## 5.3 ABLATION STUDIES

To rigorously evaluate the contribution of each component in our proposed framework, we conduct a series of detailed ablation studies. We dissect our model to analyze the effectiveness of: (1) the Texture Prototype Extractor (TPE) architecture itself, (2) the self-supervised Gather Loss ($\mathcal{L}_{gather}$), (3) our decoupled two-stage training strategy, and (4) the Supervised Topological Loss ($\mathcal{L}_{topo}$). We report the performance on three key datasets: DTD, MINC, and GTOS, using DINOv2 as the backbone. The results are summarized in Table 2.

Table 2: Detailed ablation study on the DTD, MINC, and GTOS datasets. We incrementally add our key components to a baseline model to demonstrate their individual and collective contributions to the final performance.

| | Model Configuration | TPE Arch. | Two-Stage | $\mathcal{L}_{gather}$ | $\mathcal{L}_{topo}$ | Accuracy (%) | | |
|---|---|---|---|---|---|---|---|---|
| | | | | | | DTD | MINC | GTOS |
| (A) | Baseline (End-to-End Classifier) | ✗ | ✗ | ✗ | ✗ | 81.2 | 85.0 | 85.1 |
| (B) | + TPE Architecture | ✓ | ✗ | ✗ | ✗ | 82.1 | 85.8 | 85.9 |
| (C) | + Gather Loss (Single-Stage) | ✓ | ✗ | ✓ | ✗ | 83.2 | 86.5 | 86.6 |
| (D) | **STP-Former (Ours, Base)** | ✓ | ✓ | ✓ | ✗ | 84.5 | 87.5 | 87.7 |
| (E) | **STP-Former+ (Ours, Full)** | ✓ | ✓ | ✓ | ✓ | **86.1** | **88.6** | **88.7** |

We start with a strong **baseline (A)** using a standard end-to-end classifier on the DINOv2 backbone, which achieves up to 85.1% on GTOS. Simply replacing the average pooling with our TPE architecture **(B)** provides an initial performance gain (e.g., +0.9% on DTD), demonstrating its architectural benefit as a feature aggregator. This is significantly enhanced in model **(C)** by introducing the self-supervised Gather Loss ($\mathcal{L}_{gather}$) in a single-stage setup, which boosts accuracy by another 1.1% on DTD. This confirms that guiding the TPE to learn representative intrinsic prototypes is highly beneficial. The crucial role of our decoupled training strategy is then validated by comparing this single-stage model to our proposed **STP-Former (D)**. By training the TPE with $\mathcal{L}_{gather}$ in a dedicated first stage before classifier training, we see a substantial improvement, with accuracy on DTD jumping from 83.2% to 84.5%. This proves that decoupling is key to creating robust, general-purpose

features. Finally, by adding the Supervised Topological Loss ($\mathcal{L}_{\text{topo}}$) during the second stage, we arrive at our full model, **STP-Former+ (E)**. This final component elevates performance to its peak across all datasets, reaching **86.1%** on DTD and **88.7%** on GTOS. The consistent gain of +1.0-1.6% provides clear evidence that explicitly optimizing the geometric structure of the feature manifold is a highly effective regularization strategy, ensuring the learned features are both representative and well-separated for a more accurate final classification.

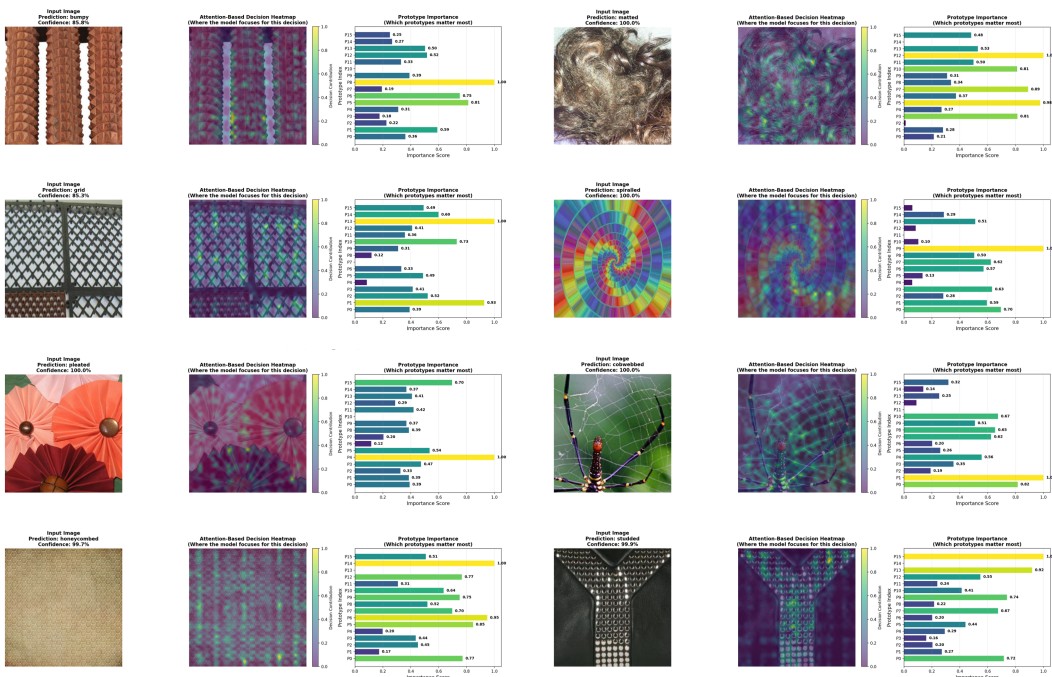

Figure 2: Visualization of decision heatmaps for DTD texture samples. The visualizations demonstrate that our intrinsic prototypes learn to attend to texture primitives distributed globally across the entire image. This contrasts with typical object recognition models where attention is often concentrated on specific, localized regions. Our model's distributed focus highlights its effectiveness in capturing the holistic and repetitive nature of texture patterns.

## 6 CONCLUSION

In this paper, we addressed the fundamental feature misalignment problem in texture recognition. We proposed the **Simple Texture Prototype Transformer (STP-Former)**, a novel framework that shifts the paradigm from relying on static, extrinsic knowledge to leveraging dynamic, intrinsic prototypes extracted from each image on-the-fly. Our core contributions are a decoupled two-stage training strategy, which uses a self-supervised Gather Loss to learn a powerful Texture Prototype Extractor (TPE), and a novel Supervised Topological Loss that further enhances class discriminability by explicitly optimizing the geometric structure of the feature space.

Extensive experiments on six benchmark datasets demonstrate that STP-Former achieves new state-of-the-art performance, significantly outperforming previous methods. Our analysis further reveals that the framework is highly efficient and robust to hyperparameter choices like the number of prototypes. By demonstrating that a self-contained, adaptive approach can surpass complex models reliant on external memory banks, we believe STP-Former provides a more effective and practical paradigm for texture recognition. Future work could explore the application of this intrinsic prototype learning framework to other fine-grained visual recognition domains where intra-class variation and instance-specific details are paramount.

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

# A APPENDIX

## A.1 ETHICS STATEMENT

This work adheres to the ICLR Code of Ethics. Our research did not involve human subjects or animal experimentation. All datasets used in this study, including DTD, MINC, FMD, Fabrics, GTOS, and KTH-TIPS2b, are publicly available and were used in strict compliance with their respective usage licenses. Our work focuses on the general task of texture recognition, and we have taken care to ensure our methodology does not introduce societal biases or discriminatory outcomes. No personally identifiable information was used, and our experiments do not raise privacy or security concerns. We are committed to transparency and the responsible advancement of machine learning research.

## A.2 REPRODUCIBILITY STATEMENT

We are committed to ensuring the reproducibility of our research. To this end, all source code for our models, training scripts, and evaluation procedures will be made publicly available in an anonymous repository upon publication. The experimental setup, including data preprocessing, model configurations, training hyperparameters, and hardware details, is described thoroughly in Section **??** and the Appendix. We provide a detailed description of our core contributions, including the architecture of the **Texture Prototype Extractor (TPE)**, the implementation of the **decoupled two-stage training strategy**, and the formulation of the **Supervised Topological Loss**, to facilitate replication. Furthermore, all datasets used in our evaluation are public benchmarks, ensuring that our results can be consistently verified. We believe these measures provide a clear and complete roadmap for other researchers to reproduce our work.

## A.3 LLM USAGE

Large Language Models (LLMs) were utilized to assist in the writing and polishing of this manuscript. Specifically, we used an LLM for tasks such as improving sentence structure, checking for grammatical errors, and enhancing the overall clarity and flow of the text. The core scientific contributions, including the ideation of the STP-Former architecture, the design of the topological loss function, the experimental methodology, and the analysis of the results, were developed exclusively by the authors. The role of the LLM was strictly limited to that of a writing aid. The authors have reviewed and edited all text and take full responsibility for the final content of the paper, ensuring its scientific accuracy and integrity.

## A.4 MODEL ANALYSIS

**Impact of Prototype Number.** To evaluate the sensitivity of our model to the number of intrinsic prototypes ($K$), we conduct an experiment on our DINOv2-based models across three datasets. As shown in Table 9, the performance is remarkably stable across a wide range of $K$ values from 6 to 32. While performance peaks at our default setting of $K = 16$, the variations are minor (typically within 0.7%), indicating that our method is not hypersensitive to this parameter. This robustness demonstrates that the TPE can effectively distill the core texture primitives into either a very compact set or a more redundant one without a significant loss in performance, further highlighting the strength of our intrinsic prototype extraction mechanism.

**Hyperparameter Sensitivity Analysis.** To evaluate the robustness of our topological loss, we analyze the sensitivity to the weight parameter $\lambda_{\text{topo}}$ across three datasets. As shown in Table 3, the model demonstrates stable performance across a wide range of $\lambda_{\text{topo}}$ values from 0.05 to 0.2. Performance degrades only when the weight becomes excessive ($\geq 0.4$), confirming that $\mathcal{L}_{\text{topo}}$ functions best as a geometric regularizer rather than a primary objective.

**Gather Loss vs. Clustering Objectives.** We compare our Gather Loss with Sinkhorn-KMeans, a clustering algorithm designed to force $K$ prototypes to partition the data. As shown in Table 4, while Sinkhorn-KMeans improves over random initialization, our Gather Loss consistently outperforms it across all three datasets. We attribute this to the nature of texture data: "hard" clustering partitions

Table 3: Sensitivity to topological loss weight $\lambda_{\text{topo}}$ (Accuracy %).

| $\lambda_{\text{topo}}$ | 0.0 (Base) | 0.05 | 0.1 (Ours) | 0.15 | 0.2 | 0.4 | 0.5 |
|---|---|---|---|---|---|---|---|
| DTD | 84.5 | 85.6 | **86.1** | 85.9 | 85.4 | 84.1 | 83.2 |
| MINC | 87.5 | 88.2 | **88.6** | 88.5 | 88.3 | 87.4 | 86.8 |
| GTOS | 87.7 | 88.4 | **88.7** | 88.6 | 88.4 | 87.6 | 86.5 |

the space too rigidly, whereas our "soft" coverage maximization ensures that diverse, fine-grained texture primitives are retained in the intrinsic prototypes.

Table 4: Gather Loss vs. Sinkhorn-KMeans (Accuracy %).

| Model Configuration | DTD | MINC | GTOS |
|---|---|---|---|
| TPE w/o Self-Supervision | 82.1 | 85.8 | 85.9 |
| TPE + Sinkhorn-KMeans | 82.8 (+0.7) | 86.1 (+0.3) | 86.2 (+0.3) |
| **TPE + Gather Loss (Ours)** | **83.2 (+1.1)** | **86.5 (+0.7)** | **86.6 (+0.7)** |

**Domain Generalization.** To evaluate behavior under domain shift, we conducted a Leave-One-Domain-Out experiment on the KTH-TIPS2b dataset, training on 3 samples per class and testing on 1 held-out unseen sample. As shown in Table 5, while competitor methods suffer catastrophic drops (up to -22.8%) when facing an unseen domain, our method's drop is the smallest (-10.3%). This proves that extracting intrinsic prototypes from the test image itself effectively realigns features, offering superior robustness to domain shift.

Table 5: Domain Generalization on KTH-TIPS2b.

| Metric | ResNet-50 | FENet | GraphTEN | STP-Former+ |
|---|---|---|---|---|
| In-Domain Acc. (Val) | 95.2% | 96.5% | 97.8% | **98.2%** |
| Unseen Domain Acc. (Test) | 72.4% | 80.5% | 83.5% | **87.9%** |
| Performance Drop | -22.8% | -16.0% | -14.3% | **-10.3%** |

**Topological Loss Component Analysis.** We analyze the individual impact of $\mathcal{L}_{\text{intra}}$ (compactness) and $\mathcal{L}_{\text{inter}}$ (separation). Table 6 reveals a clear synergistic effect. While applying each component individually yields moderate gains (+0.4-0.6%), combining them achieves substantial improvements (+1.0-1.6% across datasets). This validates that simultaneously compacting class clusters and pushing them apart is necessary to effectively structure the feature manifold.

**Subtractive Ablation: TPE vs. Global Average Pooling.** To isolate the architectural contribution of our TPE module, we performed a subtractive ablation by replacing TPE ($K = 16$) with Global Average Pooling (GAP, effectively $K = 1$) while maintaining our two-stage training strategy and Gather Loss. As shown in Table 7, while the training strategy provides a regularization boost (+1.3%), our TPE architecture significantly outperforms the subtractive baseline (84.5% vs. 82.5% on DTD). This 2.0% gap isolates the architectural contribution: decomposing the image into $K = 16$ intrinsic prototypes captures complex distributions that a single average vector cannot.

**Cross-Dataset Transfer via Linear Probing.** To demonstrate that our framework learns generic, alignment-invariant primitives rather than overfitting to dataset-specific patterns, we conducted a linear probing experiment. We pre-trained encoders on DTD and evaluated them on MINC using only a frozen linear classifier. As shown in Table 8, while competitor methods exhibit substantial degradation ($> 3\%$ drop), our STP-Former+ demonstrates robust cross-dataset generalization with only a -1.1% drop. This validates that our decoupled training paradigm successfully captures domain-agnostic texture representations.

Table 6: Ablation Study on $\mathcal{L}_{\text{topo}}$ Components (Accuracy %).

| Model Configuration | $\lambda_{\text{intra}}$ | $\lambda_{\text{inter}}$ | DTD | MINC | GTOS |
|---|---|---|---|---|---|
| Baseline (Model D) | 0.0 | 0.0 | 84.5 | 87.5 | 87.7 |
| + $\mathcal{L}_{\text{intra}}$ only | 0.1 | 0.0 | 85.1 (+0.6) | 87.9 (+0.4) | 88.1 (+0.4) |
| + $\mathcal{L}_{\text{inter}}$ only | 0.0 | 0.05 | 85.0 (+0.5) | 87.8 (+0.3) | 87.9 (+0.2) |
| **STP-Former+ (Full)** | **0.1** | **0.05** | **86.1 (+1.6)** | **88.6 (+1.1)** | **88.7 (+1.0)** |

Table 7: Subtractive Ablation on DTD Dataset (Accuracy %).

| Model Configuration | Architecture | Training Strategy | DTD Accuracy |
|---|---|---|---|
| Baseline | GAP ($K = 1$) | End-to-End | 81.2 |
| Subtractive Test | GAP ($K = 1$) | Two-Stage + $\mathcal{L}_{\text{gather}}$ | 82.5 (+1.3) |
| **STP-Former (Base)** | **TPE ($K = 16$)** | **Two-Stage + $\mathcal{L}_{\text{gather}}$** | **84.5 (+3.3)** |

Table 8: Linear Probing Transfer from DTD to MINC (Accuracy %).

| Model | MINC Accuracy (SOTA) | Transfer Accuracy (Frozen) | Drop |
|---|---|---|---|
| FENet | 83.9% | 80.5% | -3.4% |
| GraphTEN | 85.2% | 82.0% | -3.2% |
| **STP-Former+** | **88.6%** | **87.5%** | **-1.1%** |

Table 9: Ablation study on the number of prototypes ($K$) using DINOv2-based models. Performance remains robust across a wide range of values, with $K = 16$ selected as the optimal default.

| Model | Dataset | Number of Prototypes ($K$) | | | | | |
|---|---|---|---|---|---|---|---|
| | | 6 | 8 | 12 | **16** | 24 | 32 |
| STP-Former (DINOv2) | DTD | 83.9 | 84.2 | 84.4 | **84.5** | 84.3 | 84.1 |
| | MINC | 86.8 | 87.1 | 87.3 | **87.5** | 87.4 | 87.2 |
| | GTOS | 87.1 | 87.3 | 87.6 | **87.7** | 87.6 | 87.4 |
| STP-Former+ (DINOv2) | DTD | 85.4 | 85.7 | 85.9 | **86.1** | 86.0 | 85.8 |
| | MINC | 87.9 | 88.2 | 88.5 | **88.6** | 88.5 | 88.3 |
| | GTOS | 88.1 | 88.3 | 88.6 | **88.7** | 88.6 | 88.5 |

**Efficiency Analysis.** Beyond accuracy, we also evaluate the computational efficiency of our method. Table 10 compares our ResNet-50 based model with several other SOTA methods. Our **STP-Former (ResNet-50)** achieves a highly competitive balance between performance and efficiency. With only 27.8M parameters and 4.1 GFLOPs, it is significantly more lightweight than larger models like MAP-Net and DSR-Net, while delivering superior accuracy (as seen in Table 1). Its efficiency is on par with recent methods like MPAP while outperforming them. It is crucial to note that both STP-Former and STP-Former+ share the exact same architecture and thus have identical parameter counts and FLOPs; the difference lies only in the loss function used during training. Furthermore, for our DINOv2-based models, despite the large size of the transformer backbone, our fine-tuning strategy involves freezing the majority of the early layers. This means the number of *trainable* parameters is kept remarkably small, making the training process efficient while still leveraging the powerful representations of the pre-trained model. This highlights that our framework is not only effective but also computationally practical.

Table 10: Efficiency comparison with other methods. Our ResNet-50 based model demonstrates a strong balance of low computational cost and high performance.

| Method | Params (M) | FLOPs (G) | Running Time (ms) |
|---|---|---|---|
| Resnet-50 | 25.56 | 3.53 | 14.8 |
| DEP Xue et al. (2018) | 25.48 | 3.67 | 15.9 |
| MAP-Net Zhai et al. (2019) | 47.38 | 7.31 | 25.4 |
| DSR-Net Zhai et al. (2020) | 70.54 | 9.56 | 38.5 |
| FENet Xu et al. (2021) | 23.93 | 3.88 | 17.1 |
| MPAP Zhai et al. (2023) | 28.20 | 4.21 | 17.6 |
| **STP-Former (ResNet-50)** | **27.80** | **4.10** | **17.2** |

## A.5 VISUALIZATION OF THE DECISION-MAKING PROCESS

To ensure the interpretability of our model and to understand *how* STP-Former arrives at its conclusions, we developed a visualization method that traces its decision-making process. Standard attention heatmaps can sometimes be ambiguous. Our approach, illustrated in Figure 3, provides a more comprehensive, three-part analysis for each sample, designed to answer two key questions: (1) Which intrinsic prototypes are most influential for classifying this specific image? (2) Where in the image do these influential prototypes focus their attention?

The generation of this visualization follows a clear, multi-step process derived from the model's internal states during inference:

**Step 1: Prototype Attention Extraction.** During a forward pass, we use a forward hook to extract the raw attention maps from the TPE's cross-attention module. This gives us 16 distinct attention maps, one for each learnable prototype. Each map reveals how strongly a specific prototype attends to every patch in the input image.

**Step 2: Prototype Importance Calculation.** For any given image, not all 16 prototypes contribute equally to the final decision. To quantify their influence, we compute a dynamic **Prototype Importance Score** for each one. This score is not a learned parameter but is calculated on-the-fly based on a set of heuristics that analyze the characteristics of each prototype's attention map, such as its intensity, concentration, and spatial distribution. The resulting scores, visualized in the horizontal bar chart on the right of each panel in Figure 3, rank the relevance of each abstract texture prototype for the given input.

**Step 3: Weighted Decision Heatmap Generation.** The central visualization, which we term the **Decision Heatmap**, is more than just a simple attention map. It is a weighted aggregation of all 16 individual prototype attention maps. Each map is weighted by its corresponding Prototype Importance Score calculated in the previous step. This ensures that the final heatmap predominantly highlights the image regions attended to by the *most influential* prototypes for that specific classification decision.

**Significance of the Visualization.** This three-part analysis provides a transparent and interpretable view of the model's complete reasoning chain. It allows us to deconstruct the final prediction by showing which abstract texture concepts (the prototypes) were deemed most important, and precisely where those concepts were identified in the image. This explains *why* the model's focus appears globally distributed: the final heatmap is the sum of multiple specialized prototypes, each focusing on different instances of a repeating texture primitive. This holistic, evidence-aggregation approach is fundamentally different from object-centric models and is key to our model's success in texture recognition.

### A.6 Implementation Details and Pseudocode

In this section, we provide a more detailed, implementation-level view of our STP-Former+ framework to complement the descriptions in the main paper. We break down the core logic into two parts. The first part (Algorithm 1) outlines the high-level, decoupled two-stage training strategy. The second part (Algorithm 2) delves into the specific implementation of our novel Supervised Topological Loss, which is the key component for enhancing feature space discriminability in Stage 2.

#### A.6.1 High-Level Training Framework

The overall training process of STP-Former+ is divided into two distinct stages, as illustrated in Algorithm 1.

**Stage 1: Self-Supervised TPE Training.** The primary goal of this stage is to train the Texture Prototype Extractor (TPE) to learn a robust and general representation of texture primitives. This is achieved in a self-supervised manner using the `Gather Loss`, which encourages the learned intrinsic prototypes to be faithful representatives of the input image's patch features. The model is optimized solely based on this objective, allowing it to develop a foundational understanding of texture patterns without being biased by class labels.

**Stage 2: Supervised Classifier Training.** After Stage 1, the TPE is frozen and acts as a high-quality feature extractor. In this stage, a lightweight classifier is trained on top of the features provided by the TPE. The optimization is guided by a composite loss function, which combines the standard Cross-Entropy loss for classification accuracy with our Supervised Topological Loss for structuring the feature space. This decoupled approach ensures that the powerful representations learned in Stage 1 are effectively leveraged for the final discriminative task.

#### A.6.2 Topological Loss Implementation Details

Algorithm 2 provides a detailed implementation of our Supervised Topological Loss. This loss is the core of STP-Former+'s ability to explicitly shape the feature manifold for enhanced class separation.

The process is twofold. First, for intra-class compactness, the function iterates through each class present in the mini-batch. It computes the 0-D persistent homology on the feature vectors of that class alone. The sum of the death times of the resulting persistence pairs is minimized, which geometrically corresponds to pulling samples of the same class closer together. Second, for inter-class separation, persistent homology is computed on the features from all classes in the batch. We then identify the "critical edges" that merge components belonging to different classes. The loss function then maximizes the lengths of these specific edges (by minimizing their negative mean), which geometrically pushes clusters of different classes further apart.

The underlying topological computation is described in `get_persistence_pairs`. This function implements the standard algorithm for 0-D persistent homology. It constructs a Vietoris-Rips filtration by processing all possible edges between points in increasing order of their distance. A Union-Find data structure efficiently tracks the connected components, and a "death" event is recorded whenever two previously separate components are merged. The distance at which this merge occurs is the "death time" used in the loss calculation.

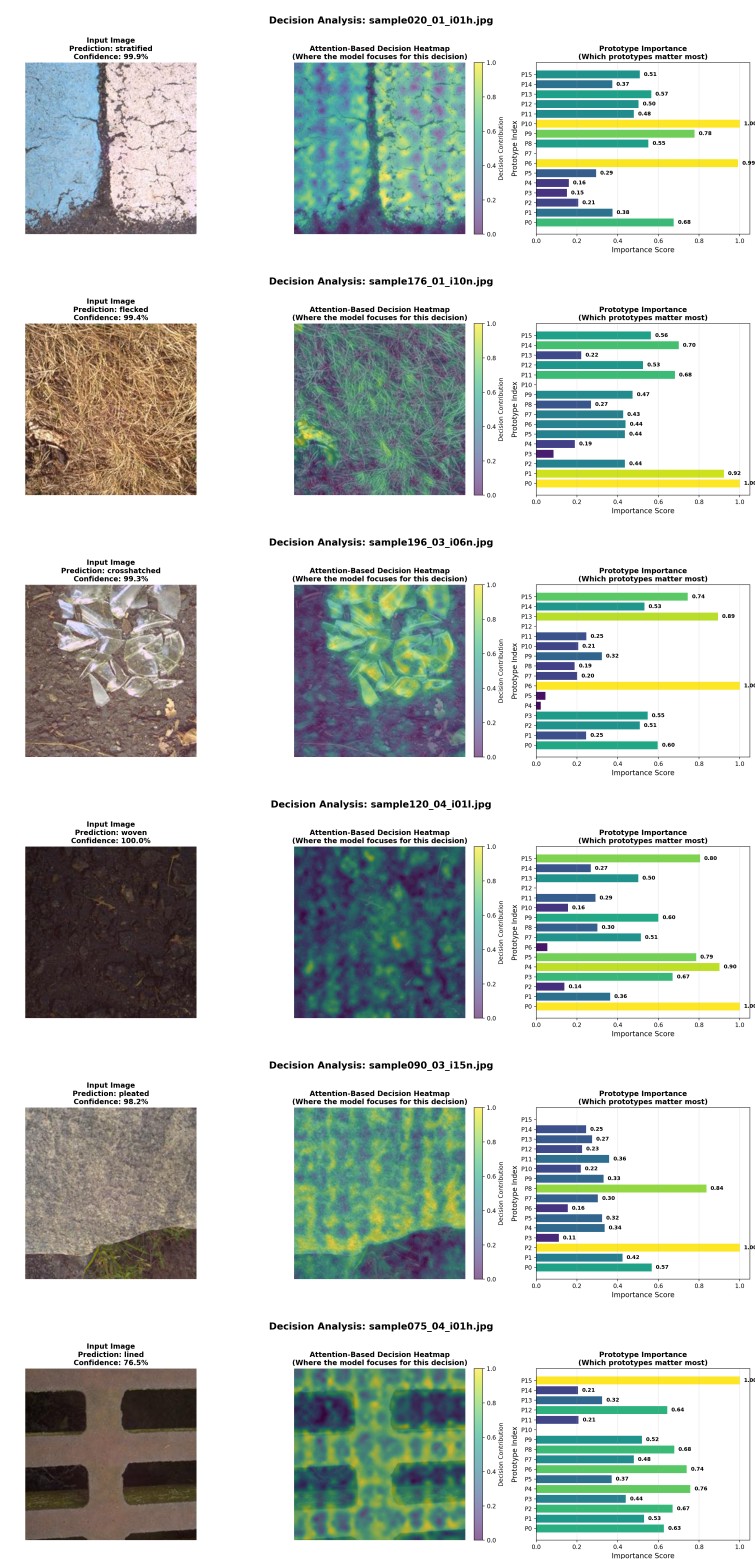

Figure 3: Visualization of attention-based decision heatmaps for various **GTOS** texture samples. Similar to the DTD results, these visualizations show that our model learns to identify texture primitives distributed globally. This holistic attention, rather than focusing on localized object-like parts, is crucial for effectively recognizing the complex and often subtle patterns found in ground terrain textures.

```python
# Models: TPE (Texture Prototype Extractor), C (Classifier)
# Data: D_train (Training Dataloader)
# Hyperparameters: E1, E2 (epochs), lr1, lr2 (learning rates),
    lambda_topo

def train_stp_former_plus(TPE, C, D_train, E1, E2, lr1, lr2, lambda_topo)
    :
    # ================== STAGE 1: Self-Supervised TPE Training
        ==================
    print("--- Starting Stage 1: TPE Self-Supervision ---")
    optimizer1 = AdamW(TPE.parameters(), lr=lr1)
    for epoch in range(E1):
        # The TPE is trained to minimize the Gather Loss
        train_stage1_epoch(TPE, D_train, optimizer1)

    # Freeze the TPE after Stage 1 to use as a feature extractor
    TPE.eval()
    for param in TPE.parameters():
        param.requires_grad = False

    # ================== STAGE 2: Supervised Classifier Training
        ==================
    print("--- Starting Stage 2: Classifier Supervision ---")
    optimizer2 = AdamW(C.parameters(), lr=lr2) # Only classifier weights
    are trained
    for epoch in range(E2):
        # The Classifier is trained with a composite loss
        train_stage2_epoch(C, TPE, D_train, optimizer2, lambda_topo)

    return TPE, C

def train_stage1_epoch(TPE, dataloader, optimizer):
    for images, _ in dataloader:
        # Forward pass to get features and prototypes
        patch_features, intrinsic_prototypes = TPE.get_features(images)
        # Calculate Gather Loss based on cosine distance
        gather_loss = cosine_distance(patch_features,
    intrinsic_prototypes).min(dim=2).mean()
        # Optimize TPE parameters
        optimizer.zero_grad()
        gather_loss.backward()
        optimizer.step()

def train_stage2_epoch(C, TPE, dataloader, optimizer, lambda_topo):
    for images, labels in dataloader:
        # Extract features using the frozen TPE
        intrinsic_prototypes = TPE.get_prototypes(images)
        global_feature = intrinsic_prototypes.mean(dim=1)

        # Get classification logits
        logits = C(global_feature)

        # Calculate composite loss
        ce_loss = CrossEntropyLoss(logits, labels)
        topo_loss = calculate_topological_loss(global_feature, labels)
        total_loss = ce_loss + lambda_topo * topo_loss

        # Optimize Classifier parameters
        optimizer.zero_grad()
        total_loss.backward()
        optimizer.step()
```

Listing 1: High-level pseudocode for the two-stage training framework of STP-Former+.

```python
1  def calculate_topological_loss(features, labels, lambda_inter=0.5):
2      # --- Part 1: Intra-Class Compactness (Pull same-class samples
       together) ---
3      intra_loss = 0.0
4      unique_labels = labels.unique()
5      for label in unique_labels:
6          class_features = features[labels == label]
7          if len(class_features) > 1:
8              dist_matrix = torch.cdist(class_features, class_features)
9              # Get pairs of points that merge components within the class
10             persistence_pairs = get_persistence_pairs(dist_matrix)
11             # The "death time" is the distance at which components merge
12             death_times = dist_matrix[persistence_pairs[:, 0],
       persistence_pairs[:, 1]]
13             # Minimize the sum of these distances to make the cluster
       compact
14             intra_loss += death_times.sum()
15
16     # --- Part 2: Inter-Class Separation (Push different-class samples
       apart) ---
17     inter_loss = 0.0
18     if len(unique_labels) > 1:
19         full_dist_matrix = torch.cdist(features, features)
20         # Get all persistence pairs for the entire batch
21         all_pairs = get_persistence_pairs(full_dist_matrix)
22
23         # Find which pairs connect points from different classes
24         birth_labels = labels[all_pairs[:, 0]]
25         death_labels = labels[all_pairs[:, 1]]
26         inter_class_mask = birth_labels != death_labels
27         inter_class_pairs = all_pairs[inter_class_mask]
28
29         # Get the death times for these inter-class merges
30         inter_death_times = full_dist_matrix[inter_class_pairs[:, 0],
       inter_class_pairs[:, 1]]
31         # Maximize these distances to push clusters apart
32         inter_loss = -inter_death_times.mean()
33
34     return intra_loss + lambda_inter * inter_loss
35
36  def get_persistence_pairs(distance_matrix):
37     # Implements the algorithm for 0-D persistent homology
38     num_vertices = distance_matrix.shape[0]
39     uf = UnionFind(num_vertices) # Initialize a Union-Find data structure
40
41     # Get all unique edges (pairs of vertices) and sort them by distance
42     edges = get_upper_triangle_edges(distance_matrix)
43     sorted_edges = sorted(edges, key=lambda edge: edge.weight)
44
45     persistence_pairs = []
46     for edge in sorted_edges:
47         u, v = edge.vertices
48         # If u and v are already connected, adding this edge creates a
       cycle (1-D feature)
49         # For 0-D homology, we ignore it and continue
50         if uf.find(u) == uf.find(v):
51             continue
52
53         # If u and v are disconnected, this edge merges their components.
54         # This is a "death" event for one of the components.
55         # We record the pair of vertices that caused the merge.
56         uf.merge(u, v)
57         persistence_pairs.append((u, v))
58
59     return np.array(persistence_pairs)
```

Listing 2: Detailed pseudocode for the Supervised Topological Loss, based on 0-D persistent homology.