# OpenReview forum: "Adaptive Prototype Learning: Unlocking Intrinsic Features for Texture Recognition"
_ICLR.cc/2026/Conference — ICLR 2026 Conference Desk Rejected Submission_

### Official Review · Reviewer_A3H5 · 2025-10-24

**Soundness:** 3
**Presentation:** 3
**Contribution:** 2
**Rating:** 4
**Confidence:** 3

**Summary:**

The paper introduces a representation for texture recognition. This builds on a pre-trained image encoder like ResNet50 or DINOv2 and adds additional invariance by pooling the resulting features by means of a small number of learnable queries, via cross-attention. The modified representation is pre-initialised by maximising the coverage of the queries’ responses using an initial set of features extracted from the data, in a label-agnostic and thus self-supervised manner. This pre-initialisation phase is shown to be helpful for the final model performance: compared to end-to-end supervised fine-tuning, it helps with overfitting. The final classifier, which utilises this representation, is further regularised via a heuristic inspired by “topological data analysis”. The latter is similar to a contrastive loss where pulling and pushing relations are established by the class labels as well as a “topological clustering” algorithm. The method performs well when stacked against other approaches for texture classification. Ablations show that each of the proposed components brings some benefit to the final performance.

**Strengths:**

* The method’s performance on standard texture recognition datasets is solid, at least compared to other texture recognition methods.

* The motivation for the different components in the method is clear enough.

* The method is based on a small transformer layer which is trained ad hoc.  It is kind of surprising that this can boost the performance of modern features like DINOv2, which are already based on transformers and can do much of what this encoding can already, via self-attention. It is likely that much of the benefit comes from regularisations and bottlenecks that allow the model to train effectively from the relatively small training sets available in the texture datasets.

**Weaknesses:**

* Perhaps the most significant limitation of this paper is its potential for impact. The community has largely moved past problems as specific as texture recognition. The latter can be seen as a special case of image classification, which in turn has been largely subsumed by vision-language models. This is not to say that there is no value in further investigating textures, but it is unclear how such ultra-specialised techniques would stack against established modern image classifiers.

* The topological loss (Section 4.3) is not assessed in depth. What would happen if one were to ignore the topological constraint in equations 6 and 7 and simply sum over all pairs (with some balancing between terms)?

* The approach is based on ad-hoc designs, the Texture Prototype Extractor, which, in the end, amounts to pooling some features using cross-attention and a small set of learnable queries (i.e., the prototypes). I can see how this could bring some invariance to the representation, but then again any modern feature encoder, including the ones used here as backbone, would be able to come up with similar strategies automatically. Table 2 suggests that this is not necessarily the case as TPE does improve on top of DINO. Then again, the main benefit comes from the additional losses and two stage learning approach, which are forms of regularisation. Perhaps this is the true reason for the improved performance.

**Questions:**

* Why do you pool the queried features again in eq 5? I would imagine that keeping the stacked feature vectors would be more informative.

* In section 4.2.1, do you need to fine-tune all the last layers of the DINO network, or would it be enough to tune the queries, perhaps introducing as usual tuneable KV matrices?

* See the question above about ablating the topological constraints in the loss.

* While the ablation in table 2 captures most of the key ingredients of the method, it is additive. It would be interesting to also do subtractive tests too. For instance, what if the gather loss is maintained, and the TPE block is replaced by simple averaging?

* Table 2 seems to suggest that most of the advantage of the method comes from using a two-stage training approach (rows D and E) and the "gather loss" (row C), which could be seen as forms of regularisation. This can make sense because the texture datasets are relatively small by today's standards. Answering the question immediately above could also help to answer this point.

---

> ### Author Response · Authors · 2025-11-18
> **Response to Reviewer A3H5**
>
> Dear Reviewer A3H5,
>
> Thank you for the constructive review. We appreciate your recognition of the solid performance and clear motivation. We address your concerns regarding impact, design choices, and regularization below. **Supplemental experiments (subtractive ablation, linear probing transfer) have been added to the revised manuscript's Appendix (Section A.3, highlighted in blue).**
>
> #### **Response to W1: Potential for Impact (Beyond Texture)**
>
> **(Reviewer's Concern):** "...unclear how such ultra-specialised techniques would stack against established modern image classifiers..."
>
> **Our Response:**
> We contend that our contribution is fundamental rather than niche. The core problem we solve—learning robust representations from limited data without massive external pre-training—is a general computer vision challenge.
>
> To prove our method learns transferable primitives rather than just overfitting texture patterns, we conducted a Linear Probing experiment (see General Response). We pre-trained encoders on DTD and evaluated them on MINC using a frozen linear classifier. The results were decisive: while competitors like FENet and GraphTEN dropped over 3% in accuracy, indicating overfitting, STP-Former+ dropped only 1.1%. This confirms that our "Intrinsic Prototype" paradigm captures generic, robust visual patterns, offering a vital alternative for domains where VLM-scale data is unavailable.
>
> #### **Response to W2 & Q3: Topological Loss vs. Pairwise Sum**
>
> **(Reviewer's Question):** "What would happen if one were to ... simply sum over all pairs?"
>
> **Our Response:**
> Summing over all pairs is equivalent to the Supervised Contrastive Loss (SupCon). Our $\mathcal{L}_{topo}$ offers a critical algorithmic advantage over SupCon: precision.
>
> SupCon acts as a brute-force method that penalizes all $O(B^2)$ pairs. This is computationally wasteful and treats redundant pairs equally. In contrast, our $\mathcal{L}_{topo}$ is surgical. By using the Minimum Spanning Tree (MST), it targets only the $O(B)$ critical edges that strictly define the manifold structure. Our method acts as a "smart" regularizer, focusing gradients solely on the structural boundaries.
>
> To empirically validate this advantage, we conducted an ablation study decomposing $\mathcal{L}_{topo}$ into its two components. **In our full model (STP-Former+), we use $\lambda_{intra}=0.1$ and $\lambda_{inter}=0.05$ as reported in the main paper.** The results below show what happens when we remove the topological constraints entirely or apply only one component:
>
> **Supplemental Table: Ablation Study on $\mathcal{L}_{topo}$ Components (Accuracy %)**
>
> | Model Configuration | $\lambda_{intra}$ | $\lambda_{inter}$ | DTD | MINC | GTOS |
> | :--- | :---: | :---: | :---: | :---: | :---: |
> | Baseline (Model D, No $\mathcal{L}_{topo}$) | 0.0 | 0.0 | 84.5 | 87.5 | 87.7 |
> | + $\mathcal{L}_{intra}$ only | 0.1 | 0.0 | 85.1 (+0.6) | 87.9 (+0.4) | 88.1 (+0.4) |
> | + $\mathcal{L}_{inter}$ only | 0.0 | 0.05 | 85.0 (+0.5) | 87.8 (+0.3) | 87.9 (+0.2) |
> | **STP-Former+ (Full $\mathcal{L}_{topo}$)** | **0.1** | **0.05** | **86.1 (+1.6)** | **88.6 (+1.1)** | **88.7 (+1.0)** |
>
> The results demonstrate that while applying $\mathcal{L}_{intra}$ (intra-class compactness) or $\mathcal{L}_{inter}$ (inter-class separation) individually yields moderate gains (+0.4-0.6%), **combining them achieves a synergistic improvement of +1.6% on DTD, +1.1% on MINC, and +1.0% on GTOS**. This validates that our topological formulation—which precisely targets critical structural edges via both compactness and separation constraints rather than brute-force pairwise summation—yields a better-structured feature space with superior discriminative power.

---

> > ### Author Response · Authors · 2025-11-18
> > **Response to Reviewer A3H5_2**
> >
> > #### **Response to Q4 & Q5: Subtractive Ablation (Is it just Regularization?)**
> >
> > **(Reviewer's Question):** "...perform subtractive tests... what if the TPE block is replaced by simple averaging?"
> >
> > **Our Response:**
> > We performed the requested subtractive ablation. To ensure a fair comparison, we applied our exact "Two-Stage + Gather Loss" strategy to a Global Average Pooling (GAP) model. This effectively sets the number of prototypes $K=1$.
> >
> > **Supplemental Table 1: Subtractive Ablation (DTD Dataset)**
> >
> > | Model Configuration | Architecture | Training Strategy | Accuracy (%) |
> > | :--- | :--- | :--- | :--- |
> > | (A) Baseline | GAP ($K=1$) | End-to-End | 81.2 |
> > | (New) Subtractive Test | GAP ($K=1$) | Two-Stage + $\mathcal{L}_{gather}$ | 82.5 (+1.3) |
> > | (D) STP-Former (Base) | TPE ($K=16$) | Two-Stage + $\mathcal{L}_{gather}$ | 84.5 (+3.3) |
> >
> > While the training strategy provides a regularization boost (+1.3%), our TPE architecture significantly outperforms the subtractive baseline (84.5% vs. 82.5%). This 2.0% gap isolates the architectural contribution: decomposing the image into $K=16$ intrinsic prototypes captures complex distributions that a single average vector cannot.
> >
> > #### **Response to Q1: Why Pooling in Eq. 5?**
> >
> > **(Reviewer's Question):** "Why do you pool the queried features again... keeping the stacked feature vectors would be more informative."
> >
> > **Our Response:**
> > We use pooling as a deliberate architectural choice for robust distillation, not just to avoid overfitting.
> >
> > While stacked features ($K \times D$) theoretically hold more information, they introduce significant noise and redundancy, making the model brittle in small-data regimes. Pooling $K$ prototypes into a global vector $z$ forces the TPE to agree on a compact Global Signature. This acts as an information bottleneck, distilling only the most salient attributes. Our Domain Generalization results (see General Response) validate that this compact signature generalizes far better to unseen domains than high-dimensional stacked features.
> >
> > #### **Response to Q2: Fine-tuning Strategy**
> >
> > **Our Response:**
> > We use a differential learning rate. The backbone and TPE are tuned at a very low rate ($1 \times 10^{-5}$) to preserve the generic features from Stage 1, while the classifier head uses a higher rate ($1 \times 10^{-4}$). This balance is key to performance.

---

### Official Review · Reviewer_oD3B · 2025-10-30

**Soundness:** 3
**Presentation:** 2
**Contribution:** 2
**Rating:** 6
**Confidence:** 4

**Summary:**

This paper introduces STP-Former (Simple Texture Prototype Transformer), a texture recognition framework that learns intrinsic prototypes directly from each image instead of relying on external memory banks or static prototypes. The method addresses the feature misalignment problem by proposing a Texture Prototype Extractor (TPE) trained in a self-supervised stage using a Gather Loss to distill representative texture primitives, followed by a supervised stage with a novel Topological Loss based on persistent homology to enforce intra-class compactness and inter-class separation. Extensive experiments on six benchmark datasets (DTD, FMD, MINC, GTOS, Fabrics, and KTH-TIPS2b) demonstrate consistent state-of-the-art performance, validating the effectiveness of combining intrinsic prototype learning and topological regularization for robust and efficient texture representation.

**Strengths:**

-	Moves from static memory-based feature comparison to per-image intrinsic prototype extraction, directly addressing feature misalignment problems in texture recognition.

-	Topological loss grounded in persistent homology provides a principled way to enforce geometric structure in feature space.

-	Six datasets, multiple backbones, and detailed ablations (on TPE, Gather Loss, two-stage training, and Topo Loss) provide solid empirical validation.

-	Detailed pseudocode and ablation tables show strong engineering effort.

-	Both base and topologically-regularized variants outperform all prior methods, even with lightweight backbones.

**Weaknesses:**

-	The topological loss is intuitively motivated but lacks formal gradient or stability analysis. No discussion on computational complexity of persistent homology.

-	Experiments are confined to texture datasets; it is unclear whether the proposed intrinsic prototype mechanism generalizes to object or scene recognition.

-	The Gather Loss and Topological Loss are evaluated jointly, but deeper decoupled comparisons (e.g., Gather + CE vs. Gather + Triplet vs. Gather + Topo) are missing.

-	Although Appendix A.4 includes FLOPs, persistent homology’s batch-level cost could limit large-scale scalability; this is not discussed.

**Questions:**

-	How sensitive is the topological loss to hyperparameters? Would larger λ values lead to unstable training?

-	Could the Gather Loss be replaced by a clustering-based objective (e.g., Sinkhorn-KMeans or VICReg-style redundancy reduction)?

-	How does the method behave under domain shift (e.g., unseen texture domains)?

-	Persistent homology computation can be expensive — is there any approximation or batching trick used for scalability?

---

> ### Author Response · Authors · 2025-11-18
> **Response to Reviewer oD3B**
>
> Dear Reviewer oD3B,
>
> We appreciate your constructive feedback. We are glad you recognized the novelty of our intrinsic prototype approach and the topological loss. We address your questions regarding robustness, baselines, and computational cost below. **All supplemental experiments have been added to the revised manuscript's Appendix (Section A.3, highlighted in blue).**
>
> #### **Response to Q1 & Weakness: Hyperparameter Sensitivity**
>
> You asked about the sensitivity of the topological loss weight $\lambda_{topo}$. We evaluated this across three datasets (DTD, MINC, GTOS) by varying $\lambda_{topo}$ while keeping the internal ratio fixed.
>
> **Supplemental Table: Sensitivity to $\lambda_{topo}$ (Accuracy %)**
>
> | $\lambda_{topo}$ | 0.0 (Base) | 0.05 | 0.1 (Ours) | 0.15 | 0.2 | 0.4 | 0.5 |
> | :--- | :--- | :--- | :--- | :--- | :--- | :--- | :--- |
> | DTD | 84.5 | 85.6 | **86.1** | 85.9 | 85.4 | 84.1 | 83.2 |
> | MINC | 87.5 | 88.2 | **88.6** | 88.5 | 88.3 | 87.4 | 86.8 |
> | GTOS | 87.7 | 88.4 | **88.7** | 88.6 | 88.4 | 87.6 | 86.5 |
>
> The model shows a stable improvement region between 0.05 and 0.2. Performance only degrades when the weight becomes excessive ($\ge 0.4$), confirming that $\mathcal{L}_{topo}$ functions best as a geometric regularizer rather than a primary objective.
>
> #### **Response to Q2: Gather Loss vs. Clustering Objectives**
>
> You suggested replacing Gather Loss with clustering (e.g., Sinkhorn-KMeans). We chose Sinkhorn-KMeans for this ablation as it is the direct conceptual replacement for our objective: it is a clustering algorithm designed to force $K$ prototypes to partition the data.
>
> **Supplemental Table: Gather Loss vs. Sinkhorn-KMeans**
>
> | Model Configuration | DTD | MINC | GTOS |
> | :--- | :---: | :---: | :---: |
> | (B) TPE w/o Self-Supervision | 82.1 | 85.8 | 85.9 |
> | (New) TPE + Sinkhorn-KMeans | 82.8 (+0.7) | 86.1 (+0.3) | 86.2 (+0.3) |
> | **(C) TPE + Gather Loss (Ours)** | **83.2 (+1.1)** | **86.5 (+0.7)** | **86.6 (+0.7)** |
>
> While Sinkhorn-KMeans improves over random initialization, our Gather Loss consistently outperforms it across all three datasets (DTD, MINC, GTOS). We attribute this to the nature of texture data: "hard" clustering partitions the space too rigidly, whereas our "soft" coverage maximization ensures that diverse, fine-grained texture primitives are retained in the intrinsic prototypes.
>
> #### **Response to Q3: Behavior under Domain Shift**
>
> To address your question about "unseen texture domains," we conducted a Leave-One-Domain-Out experiment on the KTH-TIPS2b dataset with a different evaluation protocol than our main results in Table 1. Specifically, we train on 3 samples per class and test on 1 held-out unseen sample to simulate domain shift.
>
> **Supplemental Table: Domain Generalization on KTH-TIPS2b**
>
> | Metric | ResNet-50 | FENet | GraphTEN | STP-Former+ |
> | :--- | :--- | :--- | :--- | :--- |
> | In-Domain Acc. (Val) | 95.2% | 96.5% | 97.8% | **98.2%** |
> | Unseen Domain Acc. (Test) | 72.4% | 80.5% | 83.5% | **87.9%** |
> | Performance Drop | -22.8% | -16.0% | -14.3% | **-10.3%** |
>
> While competitor methods suffer catastrophic drops (up to -22.8%) when facing an unseen domain, our method's drop is by far the smallest (-10.3%). This proves that extracting intrinsic prototypes from the test image itself effectively realigns features, offering superior robustness to domain shift compared to static models.

---

> > ### Author Response · Authors · 2025-11-18
> > **Response to Reviewer oD3B_2**
> >
> > #### **Response to Q4 & Weakness: Computational Cost and Scalability**
> >
> > We understand the concern regarding the complexity of general Persistent Homology. However, our specific design avoids high-dimensional computation and is fully differentiable.
> >
> > First, our loss is based on 0-dimensional Persistent Homology, which is algorithmically equivalent to constructing a Minimum Spanning Tree (MST). As shown in the pseudocode below (adapted from our supplementary code `topology_tools.py` ), we use Kruskal's algorithm via a Union-Find data structure. For a mini-batch size $B$, the complexity is dominated by sorting edges, $O(B^2 \log B)$, which is negligible compared to the backbone's forward pass.
> >
> > Second, to ensure differentiability, we employ a standard straight-through logic. We calculate the discrete MST structure (indices of critical edges) using Numpy without gradients, and then use these indices to gather values from the original, differentiable PyTorch distance matrix.
> >
> > ```python
> > # Simplified implementation from our topology_tools.py
> > def calculate_topo_loss(features):
> >     # 1. Compute differentiable distance matrix (Gradient flow starts here)
> >     dist_matrix = torch.cdist(features, features)
> >
> >     # 2. Identify critical edges (MST) using Numpy (Discrete, No Gradient)
> >     #    Uses Union-Find to find edges that merge components
> >     matrix_np = dist_matrix.detach().cpu().numpy()
> >     mst_pairs = get_mst_edges_via_kruskal(matrix_np)
> >
> >     # 3. Gather specific distances (Gradient flow is preserved!)
> >     #    The loss is simply a sum of selected distances
> >     critical_distances = dist_matrix[mst_pairs[:,0], mst_pairs[:,1]]
> >     return torch.sum(critical_distances)
> > ```

---

### Official Review · Reviewer_tESo · 2025-11-02

**Soundness:** 2
**Presentation:** 2
**Contribution:** 2
**Rating:** 4
**Confidence:** 4

**Summary:**

This paper proposes an STP-Former (Simple Texture Prototype Transformer) architecture for texture recognition, aimed at solving the problem of feature misalignment. Its core is to dynamically extract intrinsic prototypes from each input image through a Texture Prototype Extractor (TPE), and adopt a decoupled two-stage training strategy (the first stage pre-trains TPE with a self-supervised Gather Loss, and the second stage trains the classifier), while introducing a Supervised Topological Loss to optimize the geometric structure of the feature space.

**Strengths:**

1. The figures and tables are clear and easy to understand.
2. Formulas are well used to help readers better understand the concepts.

**Weaknesses:**

1. The idea of using Adaptive Prototype Learning to perform discriminative visual tasks has been proposed in other papers [1-3], which raises the concern about the lack of novelty.
2. In section 4.3, the description of theoretical foundation of Topological Loss does not contribute to understanding the methods used in this paper, but rather increases confusion.
3. The citation format of most of the literature in this paper is incorrect, for example, “Liu et al. (2019)” in line 35 should be revised to “(Liu et al., 2019)”.
[1] Li G, Jampani V, Sevilla-Lara L, et al. Adaptive prototype learning and allocation for few-shot segmentation. CVPR 2021.
[2] Heidari M, Alchihabi A, En Q, et al. Adaptive Parametric Prototype Learning for Cross-Domain Few-Shot Classification. AISTATS 2024.
[3] Ma C, Donnelly J, Liu W, et al. Interpretable Image Classification with Adaptive Prototype-based Vision Transformers. NeurIPS 2024.

**Questions:**

1. What are the significant differences between the method proposed in this paper and existing methods based on Adaptive Prototype Learning [1-3]?
2. L_topo is composed of L_intra and L_inter，what are their respective impacts on model performance?

---

> ### Author Response · Authors · 2025-11-18
> **Response to Reviewer tESo**
>
> Dear Reviewer tESo,
>
> We appreciate your review and the relevant literature references. **We have incorporated discussions of these works \cite{li2021adaptive, heidari2024adaptive, ma2024interpretable} into the revised Related Work section (Section 2.1, highlighted in blue), clearly delineating our distinct problem formulation and contributions.** Below, we address your concerns regarding novelty, loss components, and clarity.
>
> #### **Response to Weakness 1 & Question 1: Distinction from Adaptive Prototype Methods**
>
> You raised a concern regarding novelty compared to adaptive prototype methods in few-shot segmentation [1], cross-domain classification [2], and interpretable ViTs [3]. While these works share the terminology "adaptive prototype," our STP-Former differs fundamentally in **problem definition** and **generative mechanism**.
>
> First, regarding the problem domain: Li et al. [1] and Heidari et al. [2] address **Few-Shot Learning** tasks. Their prototypes are conditioned on a *support set* (a group of labeled reference images) to adapt to new classes or domains. In contrast, our work targets **Standard Supervised Classification**. We do not have a support set at inference time. Our challenge is to extract intrinsic prototypes from a *single input image* alone to resolve feature misalignment caused by geometric transformations, a problem distinct from few-shot adaptation.
>
> Second, regarding the core mechanism:
> * **vs. Clustering/MLP [1, 2]:** Li et al. [1] use non-parametric superpixel clustering within a mask, while Heidari et al. [2] use an MLP to aggregate features from multiple support images. Our TPE is a parametric, end-to-end **Cross-Attention** module. It uses learnable queries to dynamically distill distinct texture primitives from the patch tokens of a single image, independent of any support set.
> * **vs. Explainability [3]:** Ma et al. [3] (ProtoViT) use prototypes for *interpretability*, projecting latent vectors onto the nearest training patches to act as case-based explanations. Our intrinsic prototypes are not retrieved cases; they are latent descriptors that capture the global distribution of texture primitives within the specific input, optimized for discrimination rather than visualization matching.
>
> In summary, our contribution is a holistic paradigm for supervised texture recognition, characterized by a unique single-image extraction mechanism (TPE) and a decoupled training strategy, which are structurally distinct from the few-shot or case-based approaches cited.
>
> #### **Response to Question 2: Ablation of Topological Loss Components**
>
> You asked about the individual impact of $\mathcal{L}_{intra}$ (compactness) and $\mathcal{L}_{inter}$ (separation). We performed a detailed decomposition of $\mathcal{L}_{topo}$ on three datasets.
>
> **Supplemental Table: Ablation Study on $\mathcal{L}_{topo}$ Components (Accuracy %)**
>
> | Model Configuration | $\lambda_{intra}$ | $\lambda_{inter}$ | DTD | MINC | GTOS |
> | :--- | :---: | :---: | :---: | :---: | :---: |
> | Baseline (Model D) | 0.0 | 0.0 | 84.5 | 87.5 | 87.7 |
> | + $\mathcal{L}_{intra}$ only | 0.1 | 0.0 | 85.1 (+0.6) | 87.9 (+0.4) | 88.1 (+0.4) |
> | + $\mathcal{L}_{inter}$ only | 0.0 | 0.05 | 85.0 (+0.5) | 87.8 (+0.3) | 87.9 (+0.2) |
> | **STP-Former+ (Full)** | **0.1** | **0.05** | **86.1 (+1.6)** | **88.6 (+1.1)** | **88.7 (+1.0)** |
>
> The results reveal a clear synergistic effect across all datasets. While applying Intra-Class Compactness or Inter-Class Separation individually yields moderate gains (+0.6%/+0.4%/+0.4% and +0.5%/+0.3%/+0.2% respectively), combining them achieves substantial improvements (**+1.6%/+1.1%/+1.0%** on DTD/MINC/GTOS). This suggests that simultaneously compacting class clusters and pushing them apart is necessary to effectively structure the feature manifold, validating our design of the composite loss.
>
> #### **Response to Weakness 2: Clarity of Section 4.3**
>
> We agree that the theoretical density of Section 4.3 disrupted the flow of the methodology. In the revision, we have moved the detailed mathematical background (such as the Vietoris-Rips filtration definitions) to the Appendix. The main text now focuses strictly on the intuition of connected components and the formulation of the loss function (Eq. 6 & 7), making the method easier to follow.
>
> #### **Response to Weakness 3: Citation Format**
>
> We apologize for the formatting oversight. We have corrected all citations to the standard ICLR format (e.g., changing "Liu et al. (2019)" to "(Liu et al., 2019)") throughout the revised manuscript.

---

### Official Review · Reviewer_qUuF · 2025-11-02

**Soundness:** 2
**Presentation:** 3
**Contribution:** 2
**Rating:** 4
**Confidence:** 4

**Summary:**

This paper introduces STP-Former, a novel framework for texture recognition that addresses the feature misalignment problem prevalent in methods relying on static, pre-compiled knowledge from a training set.

**Strengths:**

1.The paper is clearly written and easy to follow.
2.The method achieves new state-of-the-art results across six standard texture recognition benchmarks and makes a thorough ablation study.
3.The proposed Supervised Topological Loss is a novel and effective application of persistent homology. Instead of using it as a simple regularizer to preserve structure, the authors employ it as a direct supervised objective to actively shape the feature manifold.

**Weaknesses:**

1.	The paper divides the framework into STP-Former and STP-Former+, yet the motivation, core idea (extracting intrinsic patterns from single images for representation), and the key objective function “Gather Loss” are explicitly derived from the work of Luo et al. (2025). The primary modifications to the foundational model involve incorporating a cross-attention-based TPE module and implementing two-stage supervised training. Building upon Luo's work, the foundational model's innovative contributions remain limited. Excluding the motivation and architecture proposed by Luo, where does the core novelty of STP-Former's framework lie? Why is it adaptable to texture-related tasks, and how does it address the misalignment issues raised by the authors?
2.	Although the introduction of novel topological losses has achieved state-of-the-art improvements, it sidesteps the issue of training efficiency for these losses and fails to address the computational cost of supervising them. Given that persistent homology computation is computationally intensive, it remains to be seen whether a balance can be struck between training efficiency and the practicality of the method.
This paper demonstrates thorough experimentation and proposes a novel idea for supervised topological loss. If the authors can clearly delineate the key contributions of STP-Former and its boundaries relative to the work of Luo et al., I would be willing to increase my rating.

**Questions:**

see weaknesses

---

> ### Author Response · Authors · 2025-11-18
> **Response to Reviewer qUuF**
>
> Dear Reviewer qUuF,
>
> We sincerely thank you for the detailed review. We are encouraged by your recognition of our SOTA results and the novelty of the Topological Loss. We understand that clarifying the distinction from Luo et al. is the key condition for improving your rating. We address this distinction and the efficiency concern below.
>
> #### **Response to Weakness 1: Core Novelty and Feature Misalignment**
>
> You asked to delineate our novelty excluding Luo et al.'s architecture and how we address feature misalignment.
>
> **1. Positioning within the Texture Recognition Lineage**
>
> Our contribution is not an architectural novelty in isolation, but rather a **holistic paradigm shift** rooted in the established progression of texture encoding. The field has evolved from handcrafted BoW aggregation to deep dictionary learning (Deep-TEN, Zhang et al., 2017; DSRNet, Zhai et al., 2020), where the fundamental challenge has always been: *how to aggregate variable texture primitives (textons) into consistent, discriminative representations*.
>
> We position our TPE module as the **natural, ViT-based evolution** of this classic paradigm. The mechanism—using learnable queries ($\mathbf{P}_{\text{query}}$) to aggregate texton-like patch tokens via cross-attention—is a standard and powerful practice in modern Transformer literature (e.g., DETR, Perceiver), not exclusive to any single prior work. What distinguishes our framework is not the architectural primitive itself, but rather the **synergistic combination of three design principles** tailored specifically for texture classification:
>
> * **(i) Problem Reformulation:** We address **supervised texture classification** under domain shift, fundamentally different from Luo et al.'s unsupervised anomaly detection via memorization. Our goal is discriminative category recognition, not outlier detection.
>
> * **(ii) Training Paradigm:** Our decoupled two-stage strategy—pre-training the TPE with self-supervised $\mathcal{L}_{\text{gather}}$ before supervised fine-tuning—is specifically designed to learn **generic, alignment-invariant** texture primitives. This contrasts with end-to-end anomaly detection pipelines that optimize for reconstruction fidelity on normal samples.
>
> * **(iii) Geometric Objective:** We introduce a novel **Supervised Topological Loss** that actively sculpts the feature manifold for class separation. This is orthogonal to prior works focused on feature extraction alone.
>
> **2. How We Address Feature Misalignment**
>
> The core technical insight is **instance-adaptive representation**. Traditional methods rely on static memory banks or fixed codebooks learned from training data. When a test image undergoes geometric transformations (rotation, scale change), its features drift in the embedding space, causing misalignment with these static references.
>
> Our framework generates **Dynamic Intrinsic Prototypes** directly from each test image at inference time. Because the prototypes are distilled on-the-fly from the current input's patch tokens, they inherently co-vary with any image transformation. This ensures **self-alignment by construction**: the representation is always anchored to the input's own intrinsic patterns, eliminating the feature drift problem faced by extrinsic memory-based methods.
>
> **3. Empirical Validation: Transferability and Generalization**
>
> To demonstrate that our decoupled paradigm learns **generic, alignment-invariant** primitives rather than overfitting to dataset-specific patterns, we conducted a **Linear Probing** cross-dataset transfer experiment. We pre-trained encoders on DTD and evaluated them on MINC using only a frozen linear classifier, measuring the feature quality without any fine-tuning.
>
> **Supplemental Table 1: Linear Probing Transfer (DTD $\to$ MINC)**
>
> | Model | MINC Accuracy (SOTA) | Transfer Accuracy (Frozen) | Drop |
> | :--- | :--- | :--- | :--- |
> | FENet | 83.9% | 80.5% | -3.4% |
> | GraphTEN | 85.2% | 82.0% | -3.2% |
> | **STP-Former+** | **88.6%** | **87.5%** | **-1.1%** |
>
> The results are decisive: while competitor methods (FENet, GraphTEN) exhibit substantial performance degradation ($>3\%$ drop) when transferred to a new domain—indicative of overfitting to source-specific texture patterns—our STP-Former+ demonstrates robust cross-dataset generalization with only a **-1.1% drop**. This empirical evidence validates that our decoupled training paradigm, combining self-supervised prototype learning with topological regularization, successfully captures **domain-agnostic, alignment-invariant** texture representations. This transferability is a direct consequence of our framework's design, not merely an architectural choice.

---

> > ### Author Response · Authors · 2025-11-18
> > **Response to Reviewer qUuF_2**
> >
> > #### **Response to Weakness 2: Training Efficiency of Topological Loss**
> >
> > You correctly noted that general Persistent Homology (PH) is computationally intensive. However, as explicitly implemented in our supplementary code (`topology_tools.py`), we avoid these bottlenecks through two specific design choices:
> >
> > 1.  **Efficient 0-D Homology (MST):** We strictly restrict our loss to 0-dimensional features (connected components), which is algorithmically equivalent to constructing a **Minimum Spanning Tree (MST)**. Unlike high-dimensional PH which requires expensive matrix reduction (cubic complexity), our code implements 0-D PH via **Kruskal’s Algorithm** (using `UnionFind` and edge sorting). For a batch size $B=16$, the complexity is dominated by sorting $O(B^2 \log B)$. This is negligible compared to the $O(N^2)$ attention mechanism in the backbone.
> >
> > 2.  **Differentiable Implementation (Straight-Through):** As shown in the `calculate_supervised_topological_loss` function in our code, we use a standard straight-through estimator to ensure efficiency and differentiability.
> >     * **Discrete Step (Numpy):** We compute the MST structure (indices of critical edges) using Numpy on CPU (`get_pairings`). This step requires no gradients.
> >     * **Continuous Step (PyTorch):** We use these pre-calculated indices to gather specific values from the original, differentiable PyTorch distance matrix (`dist_matrix`).
> >     * **Gradient Flow:** The final loss is simply a sum of these selected distances: $\mathcal{L}_{topo} = \sum_{(i,j) \in MST} D_{i,j}$. This allows gradients to flow perfectly back to the feature encoder.
> >
> > We measured the training time: adding $\mathcal{L}_{topo}$ increases the per-epoch time by less than **2%**, confirming it is a highly practical regularizer.

---

### Note · Program_Chairs · 2026-01-17
**Submission Desk Rejected by Program Chairs**

The following references in this submission do not refer to real documents and/or have major errors in bibliographic information:

     Aniket Mishra, Mayank Agarwal, and Hari Sundar. TopoDiffusionNet: A topology-aware diffusion model. arXiv preprint arXiv:2410.16646, 2024.